# Toll-Like Receptor Expression Profiles in Koala (*Phascolarctos cinereus*) Peripheral Blood Mononuclear Cells Infected with Multiple KoRV Subtypes

**DOI:** 10.3390/ani11040983

**Published:** 2021-04-01

**Authors:** Mohammad Enamul Hoque Kayesh, Md Abul Hashem, Kyoko Tsukiyama-Kohara

**Affiliations:** 1Transboundary Animal Diseases Centre, Joint Faculty of Veterinary Medicine, Kagoshima University, Kagoshima 890-0065, Japan; mehkayesh@yahoo.com (M.E.H.K.); mdhashem29@yahoo.com (M.A.H.); 2Department of Microbiology and Public Health, Faculty of Animal Science and Veterinary Medicine, Patuakhali Science and Technology University, Barishal 8210, Bangladesh; 3Department of Health, Chattogram City Corporation, Chattogram 4000, Bangladesh; 4Laboratory of Animal Hygiene, Joint Faculty of Veterinary Medicine, Kagoshima University, Kagoshima 890-0065, Japan

**Keywords:** koala, toll-like receptors, koala retrovirus, immune response

## Abstract

**Simple Summary:**

Koala retrovirus (KoRV) is a major pathogen of koala. Toll-like receptors (TLRs) are important innate immune component that are evolutionary conserved and play a crucial role in the early defense against invading pathogens. The expression profile of TLRs in KoRV infection in koalas is not characterized yet. Therefore, in this study, we characterized TLR expression patterns in koalas infected with KoRV-A only vs. KoRV-A with KoRV-B and/or -C. Using qRT-PCR, we measured TLR2–10 and TLR13 mRNA expression in peripheral blood mononuclear cells (PBMCs) and/or tissues from captive koalas in Japanese zoos. We observed variations in TLR expression in koalas with a range of subtype infection profiles (KoRV-A only vs. KoRV-A with KoRV-B and/or -C). The findings of this study might improve our current understanding of koala’s immune response to KoRV infection.

**Abstract:**

Toll-like receptors (TLRs), evolutionarily conserved pattern recognition receptors, play an important role in innate immunity by recognizing microbial pathogen-associated molecular patterns. Koala retrovirus (KoRV), a major koala pathogen, exists in both endogenous (KoRV-A) and exogenous forms (KoRV-B to J). However, the expression profile of TLRs in koalas infected with KoRV-A and other subtypes is yet to characterize. Here, we investigated TLR expression profiles in koalas with a range of subtype infection profiles (KoRV-A only vs. KoRV-A with KoRV-B and/or -C). To this end, we cloned partial sequences for TLRs (TLR2–10 and TLR13), developed real-time PCR assays, and determined TLRs mRNA expression patterns in koala PBMCs and/or tissues. All the reported TLRs for koala were expressed in PBMCs, and variations in TLR expression were observed in koalas infected with exogenous subtypes (KoRV-B and KoRV-C) compared to the endogenous subtype (KoRV-A) only, which indicates the implications of TLRs in KoRV infection. TLRs were also found to be differentially expressed in koala tissues. This is the first report of TLR expression profiles in koala, which provides insights into koala’s immune response to KoRV infection that could be utilized for the future exploitation of TLR modulators in the maintenance of koala health.

## 1. Introduction

Koala (*Phascolarctos cinereus*), an iconic marsupial species of Australia, is listed as vulnerable on the International Union for Conservation of Nature (IUCN) ‘red list’ of threatened species [1]. The decline of the koala population is associated with many factors, including habitat loss, hunting, diseases, bushfire, and drought [2], of which the most interesting feature for virologists is koala retrovirus (KoRV) [3]. KoRV appears to be one of the major infectious agents in koala populations both in the wild and in captivity, and it has been implicated in neoplasia and chlamydial diseases [4,5,6,7,8,9,10,11,12]. Under the family Retroviridae and the genus *Gammaretrovirus*, KoRV is a positive-sense, single-stranded RNA virus with a genome of 8.4 kb containing gag, pol, and env genes, and long terminal repeats (LTRs) at the 5′- and 3′- ends [5,13,14]. KoRV is closely related to the gibbon ape leukemia virus (GALV), feline leukemia virus (FeLV), and porcine endogenous retrovirus (PERV) [8,13,15]. KoRV exists in both endogenous [13] and exogenous forms [7,16,17,18]. KoRV endogenization is a relatively recent event, which began not more than 49,000 years ago [19] and is presumed ongoing. To date, 10 KoRV subtypes have been reported, including KoRV-A, an endogenous form of KoRV, and nine other exogenous subtypes (KoRV-B to J) [20]. The subtypes KoRV-A and KoRV-B are most extensively characterized [3,21].

The innate immune response is critical and is considered the first line of immune defense in limiting many viral infections [22]. Viral components such as nucleic acids and proteins are sensed by different pattern recognition receptors (PRRs), including Toll-like receptors (TLRs), RIG-I-like receptors, and NOD-like receptors [23]. TLRs are the key components of innate immunity involved in early interaction with invading microorganisms by detecting pathogen-associated molecular patterns (PAMPs) present on pathogens [24], and play an important role in initiating the innate immune response and shaping adaptive immune responses that limit spreading infection [25,26]. TLRs are members of type I transmembrane proteins and have a conserved structure with an N-terminal ectodomain consisting of leucine-rich repeats, a single transmembrane domain, and a cytosolic TIR domain [27]. TLRs are encoded by a range of TLR genes that are evolutionarily conserved [28]. All organisms appear to encode a certain number of TLRs, with humans encoding 10 TLRs (TLR1–10), while mice encoding 12 TLRs [24,27]. Ten TLRs (TLR1–10) have also been reported in the gray short-tailed opossum (*Monodelphis domestica*) [29] and Tasmanian devil (*Sarcophilus harrisii*) [30]. In koalas, a recent study identified TLR2–10, TLR13, and 40 SNPs in TLR genes [31]. Previous studies have demonstrated that the innate and adaptive immune system may play an important role in endogenous retrovirus control in mice [32,33]. However, our understanding of TLRs in koala immunity remains largely unknown. With the discovery of multiple KoRV subtypes, much epidemiological research and comparative studies of its immunomodulatory effects in populations are of great importance. Moreover, KoRV subtypes appear to share a sequence for p15E, the transmembrane envelope protein, which contains an immunosuppressive domain, and is highly conserved in different retroviruses [34,35]. In addition, different preclinical and clinical studies have demonstrated that purified TLR agonists could be used as adjuvants in vaccination to enhance adaptive response [36], which could also be investigated in koalas where the antibody response is poor or lacking [37]. Therefore, a proper understanding of how TLRs respond to KoRV infection might be critical for using TLR modulators and for the development of an effective vaccine against multiple KoRV subtypes. In addition, it is also important to investigate whether koalas have a uniform or varied TLR response to the different subtypes of KoRV. As the role of TLRs in KoRV infection and/or its subtype’s differences is yet to be characterized, in this study, we characterized TLR expression profiles in koala peripheral blood mononuclear cells (PBMCs) infected with KoRV-A or KoRV-A with KoRV-B and/or -C.

## 2. Materials and Methods

### 2.1. Sample Collection

In this study, we targeted a total of 11 captive koalas (three males, eight females; age range: 6 months to 12 years) maintained in an air conditioning system (23–25 °C) in two Japanese zoos (Hirakawa Zoological Park, Kagoshima, Japan, and Kobe Oji Zoo, Kobe, Japan) to investigate TLR expression patterns in PBMCs infected with KoRV-A or KoRV-A with KoRV-B and/or -C. Tissues were collected from two koalas (KM and Joey), including one adult female koala (KM) that died at a later point in the study period [38], and its bone marrow was included for tissue expression analysis of TLRs, together with spleen and lung tissue from a joey reported in a previous study [18]. Koala whole blood samples were collected from 10 koalas (KJ, KY, KYB, KM, K1, K2, H8, KS, H7, H9) using heparin or EDTA by venipuncture between January 2019 and November 2020. The infection status of koalas was confirmed for three KoRV subtypes, KoRV-A (the endogenous subtype), and KoRV-B and KoRV-C (exogenous subtypes). A workflow of the study has been indicated in Figure 1. This study was performed following the protocols of the Institutional Animal Care and Use Committee of the Joint Faculty of Veterinary Medicine, Kagoshima University, Japan.

### 2.2. Hematological Examination

The health of the koalas was examined, and hematological examination was performed using standard protocols for determining white blood cell (WBC), red blood cell (RBC), hemoglobin level, packed cell volume (PCV), mean corpuscular volume (MCV), and mean corpuscular hemoglobin concentration (MCHC).

### 2.3. Extraction of Genomic DNA from PBMCs

Genomic DNA (gDNA) was extracted from EDTA-treated koala whole blood samples using the Wizard Genomic DNA Purification Kit (Promega, Madison, WI, USA) according to the manufacturer’s instructions. The concentration and purity of each extracted DNA were determined using a Nanodrop Spectrophotometer ND-1000 (Thermo Fisher Scientific Co., Waltham, MA, USA). The extracted gDNA was used as a template in PCR analysis for KoRV provirus detection, and for KoRV subtyping. KoRV provirus was detected by PCR analysis using KoRV pol gene-specific primers, pol F (5′-CCTTGGACCACCAAGAGACTTTTGA-3′) and pol R (5′-TCAAATCTTGGACTGGCCGA-3′), as described previously [14]. KoRV subtypes were confirmed for KoRV-A, KoRV-B, and KoRV-C using subtype-specific primers targeting the KoRV envelope gene (Table 1) [7,9,18]. PCR conditions have been described previously [38].

### 2.4. Extraction of RNA from PBMCs and Koala Tissues

Using the RNeasy Plus Mini Kit (QIAGEN, Hilden, Germany), total RNA was extracted from the isolated PBMCs according to the manufacturer’s instructions. PBMCs were isolated from whole blood samples as described previously [39] with some modifications [40]. Tissues (spleen and lung) previously collected from a six-month-old dead joey [18] were used in this study for further analysis. Bone marrow that was collected from an adult koala (KM) with leukemia, which died during the study period [38], was also included in the expression analysis. Tissue samples were kept at −80 °C until RNA extraction, and total RNA was extracted from tissues using RNeasy Plus Mini Kit (QIAGEN), according to the manufacturer’s instructions. The concentration and purity of the extracted RNA samples were confirmed using a NanoDrop ND-1000 spectrophotometer (Thermo Fisher Scientific Co.). The extracted RNA samples were kept at −80 °C until use.

### 2.5. Polymerase Chain Reaction (PCR)

A PCR assay was performed to clone the partial sequences of koala TLR2–10 and TLR13 using koala cDNA (prepared from total RNA) as the template, with koala TLR-specific primers (Table 1). The PCR conditions included denaturation at 98 °C for 2 min, 35 cycles at 98 °C for 30 s, annealing at 65 °C for 15 s, extension at 72 °C for 1 min, and a final extension at 72 °C for 5 min. Partial TLR gene PCR fragments were subcloned into pCR-Blunt II TOPO (Thermo Fisher Scientific Co.) and sequenced. Standards prepared from the prequantified plasmids containing the target gene sequence were used for gene expression analysis by quantitative reverse transcription-PCR (qRT-PCR).

### 2.6. Measurement of Gene Expression by qRT-PCR

Koala TLRs (TLR2–10, TLR13) mRNA expression levels were measured in koala PBMCs and/or tissues, using one-step qRT-PCR with Brilliant III Ultra-Fast SYBR Green qRT-PCR Master Mix (Agilent Technologies, Santa Clara, CA) according to the manufacturer’s instructions. Primers used for amplification of the TLR genes were designed using Primer-Blast (https://www.ncbi.nlm.nih.gov/tools/primer-blast/ accessed on 12 July 2020) based on the target sequences available in NCBI GenBank as shown in Table 1, and primers for amplification of the reference gene (koala beta actin) are already described [14]. The cycling conditions comprised reverse transcription at 50 °C for 10 min, initial denaturation at 95 °C for 3 min, and 40 cycles of 95 °C for 5 s and 60 °C for 10 s. Each reaction was performed in duplicate in a 20 μL volume in a 96 micro-well plate using the CFX Connect Real-Time PCR Detection System (Bio-Rad, Hercules, CA, USA). Each reaction included a no-template control and a standard curve for each gene. The specificity of each PCR reaction was confirmed by melt curve analysis. Koala beta actin was used as an endogenous control for normalization of the results.

### 2.7. Statistical Analysis

Student’s *t*-test was performed using GraphPad software for statistical analysis. *p*-Values < 0.05 were considered statistically significant. Analyses were performed to compare TLR expression based on KoRV infection profiles for KoRV-A, KoRV-B, and KoRV-C. Data are presented as mean ± standard deviation (SD).

## 3. Results

### 3.1. Infection Status of Koalas

All koalas used in this study were positive for KoRV provirus in PCR analysis with KoRV pol gene-specific primers targeting gDNA extracted from PBMCs. The infection status of each animal for the KoRV subtype was confirmed by genotyping PCR and sequencing (Table 2). The infection status of KoRV-A, KoRV-B, and KoRV-C subtypes in the study populations had been determined in previous studies [12,18,38], except for KoRV-C in koalas K1 and K2. In PCR analysis, K1 and K2 were found to be negative for the KoRV-C subtype (Table 2). All koalas used in this study were positive for KoRV-A, the endogenous subtype. Three koalas (KM, K1, and K2) were positive for KoRV-B, one koala (H8) was positive for KoRV-C, and three koalas (KS, H7, and H9) were positive for both KoRV-B and KoRV-C.

### 3.2. Hematological Examination

The hematology data for the koalas are presented in Table 3. Blood parameters were within normal ranges for seven (KJ, KY, KYB, K2, H8, H7, and H9) of the ten koalas, which were designated as “healthy”. Three koalas (KM, K1, and KS) were found to be leukemic (Table 3).

### 3.3. Expression Patterns of TLRs mRNA in Koala PBMCs

To evaluate variations in immune response according to KoRV subtype infection status, we measured the expression levels of TLRs in koala PBMCs. TLR expression analysis at the mRNA level in koala PBMCs revealed that TLRs, including TLR2–10 and TLR13 were expressed in koala PBMCs (Figure 2A–J). TLR2, TLR3, TLR9, and TLR13 expression patterns showed no significant variations in mRNA expression in koala PBMCs with a range of subtype infection profiles (KoRV-A only vs. KoRV-A with KoRV-B and/or -C) (Figure 2A,B,H,J). The mRNA expression levels of TLR4–7 and TLR10 were significantly higher in KoRV-B-and KoRV-C-positive koalas than in those with endogenous infection only (KoRV-A; KJ, KY, and KYB) (Figure 2C–F,I). In koala H8 (KoRV-B-negative and KoRV-C-positive), TLR-7, -8, and -10 expressions were markedly higher than in koalas with endogenous infection only (KoRV-A) (Figure 2F,H,I).

### 3.4. Expression Patterns of TLR mRNAs in Koala Tissues

To explore the expression patterns of TLRs in koala tissues, we measured TLR2–10 and TLR13 mRNA expression patterns in the spleen and lung tissues of a 6-months-old dead joey that was positive for KoRV-A and KoRV-C in our previous study [18]. We also measured TLR2–TLR10 and TLR13 mRNA expression levels in the bone marrow of adult koala (KM) with leukemia that died during the study period [38]. The joey showed differential expression of TLR mRNA in the spleen and lung tissues, TLR7 was found to be highly expressed in both the spleen and lung, whereas TLR13 and TLR5 were found to be least expressed in the spleen and lung, respectively (Figure 3A,B). Among the TLRs measured, TLR7 was found to be highly expressed in the bone marrow of the adult koala that died with leukemia; however, TLR13 was undetectable in the bone marrow (with 4.15 nanogram RNA) (Figure 3C).

## 4. Discussion

TLRs play a vital role in host defense against microbial infections. To the authors’ knowledge, this is the first study to profile TLR expression in koala PBMCs infected with an endogenous KoRV-A subtype, or co-infected with KoRV-B and/or KoRV-C. To this end, we characterized expression patterns of TLR2–10 and TLR13 mRNA in koala PBMCs, based on KoRV subtypes (KoRV-A, KoRV-B, and KoRV-C). In addition, we investigated the expression patterns of these key innate immune molecules in tissues from an adult koala and a joey to gain insight into the relative TLR response in KoRV infection.

It is now widely believed that PRRs of the innate immune system, particularly TLRs, are involved in modulating and shaping humoral and cellular immune responses [41]. At present, no licensed KoRV vaccine is available; however, researchers are working on KoRV vaccine development [21,42,43,44,45,46]. A proper understanding of koala’s immune response against KoRV is a key step in therapeutic and prophylactic interventions for better conservation and management strategies to protect koalas. In a recent study, a recombinant envelope protein-based KoRV vaccine was found to elicit anti-KoRV IgG and decreased viral loads in vaccinated koalas [43]. In a subsequent study by the same group, a recombinant KoRV Env protein vaccine combined with a Tri-adjuvant was found to significantly increase circulating neutralizing anti-KoRV IgG antibodies, with cross-reactivity against multiple KoRV-subtypes [45]. However, Fiebig et al., in a previous study with a limited number of koala populations, observed a lack of antibody response against KoRV infection [37]. In this study, we observed modulations of TLRs based on KoRV subtype profiles, which are suggestive of the implications of these molecules in the immune response in KoRV infection. The TLR expression profile of koala PBMCs infected with KoRV-A and KoRV-B along with leukemia was not identical, where we observed a mixed response with an overall increased expression of TLRs in koala KM, and a decreased expression in koala K1. These dissimilarities in TLR expression may indicate the active involvement of TLRs in KoRV infection. However, further investigation of TLR response in larger koala populations, including KoRV-negative koalas as well as koalas infected with multiple KoRV subtypes is warranted to gain further insight into the future exploitation of TLR modulators [47] in koala health. Moreover, retroviruses represent important human pathogens, including human immunodeficiency virus (HIV) and human T cell leukemia viruses 1 and 2 (HTLV1 and HTLV2) [48]. Numerous retroviral pathogens have also been identified in many vertebrates [49]. Retroviruses can be detected by several TLRs, which may contribute to the anti-retroviral response [50]. The suitability of TLR agonists as therapeutic tools in retroviral infections are underway [50]. Although TLRs are believed as key regulators of the anti-retroviral immune response, much remains to be learnt about the mechanisms by which TLR stimulation can inhibit, or enhance, retroviral infections. In addition, TLR response may act as a double-edged sword for the host in protecting against pathogens or induction of immune-mediated pathological consequences [51,52,53]. However, higher expression of certain TLRs in multiple KoRV infections, and their association with viral loads and pathogenesis remain to be investigated. The expression of TLRs was also investigated in tissue from a dead joey infected with KoRV-A and KoRV-C, and an adult KoRV-A- and B-positive koala that died during the study period was found leukemic. We observed differential expression patterns of TLRs in the spleen, lung, and bone marrow. However, TLR7 was found to be highly expressed in all the investigated tissues, which may indicate the active implication of TLR7 in these tissues in KoRV infection.

## 5. Conclusions

This is the first study to demonstrate TLR expression profiles and differences in TLR expression patterns in koala PBMCs with a range of subtype infection profiles (KoRV-A only vs. KoRV-A with KoRV-B and/or -C). We can clearly observe the variations in TLR expression in koala PBMC and tissues based on KoRV subtype differences, which imply the active role of TLRs in KoRV infection. Further investigations of TLR expression patterns with larger koala populations in a wider range of tissues is warranted to better understand the immune response that will facilitate prophylactic and therapeutic intervention strategies using TLR modulators for this iconic marsupial species.

## Figures and Tables

**Figure 1 animals-11-00983-f001:**
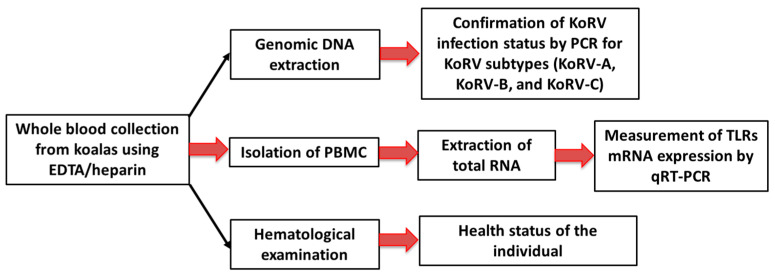
A schematic representation of the workflow of the study.

**Figure 2 animals-11-00983-f002:**
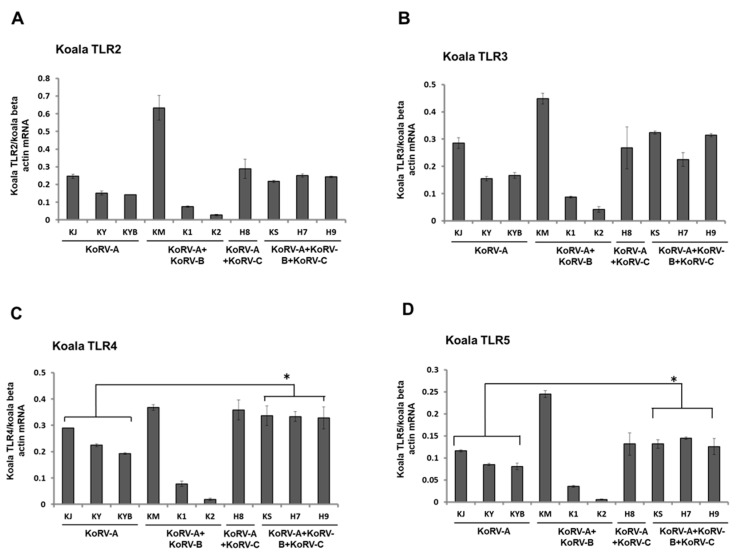
Expression profile of TLR mRNAs in koala peripheral blood mononuclear cells (PBMCs). Expression patterns of TLR2 (**A**), TLR3 (**B**), TLR4 (**C**), TLR5 (**D**), TLR6 (**E**), TLR7 (**F**), TLR8 (**G**), TLR9 (**H**), TLR10 (**I**), and TLR13 (**J**) mRNAs are indicated in koala PBMCs infected with different KoRV subtypes. Koala beta actin mRNA was used for normalization of the transcript levels of the TLRs. Statistical significance was calculated using the Student’s *t*-test and indicated by asterisks (* *p* <  0.05, ** *p* < 0.01). Data are presented as mean ± SD (*n* = 2). KoRV-A positive only koalas were considered as the control.

**Figure 3 animals-11-00983-f003:**
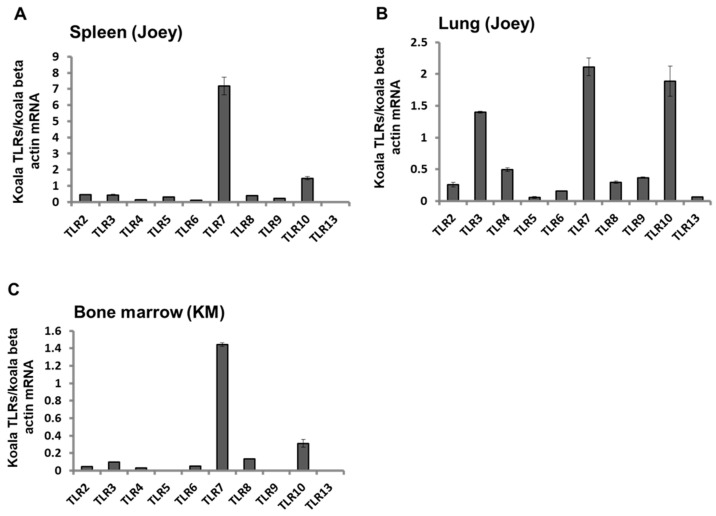
Expression profile of TLR mRNAs in koala tissues. Expression patterns of TLR2–TLR10, and TLR13 mRNAs in the spleen (**A**), and lung (**B**) of a dead joey, and bone marrow of KM (**C**) are shown. The transcript levels were normalized against koala beta actin mRNA. Data are presented as mean ± SD (*n* = 2).

**Table 1 animals-11-00983-t001:** Primers used for the detection of Koala retrovirus (KoRV) (subtypes) and toll-like receptors (TLRs) gene expression analysis.

Gene	Forward (5′ to 3′)	Reverse (5′ to 3′)	Product Size	Reference or GenBank Accession Numbers
Pol (KoRV)	CCTTGGACCACCAAGAGACTTTTGA	TCAAATCTTGGACTGGCCGA	523	[14]
Env (KoRV-A)	TCCTGGGAACTGGAAAAGAC	GGGTTCCCCAAGTGATCTG	321	[9]
Env (KoRV-B)	TCCTGGGAACTGGAAAAGAC	GGCGCAGACTGTTGAGATTC	271	[7]
Env (KoRV-C)	TCCTGGGAACTGGAAAAGAC	AAGGCTGGTCCCGCGAAGGT	290	[18]
Beta actin (Koala)	AGATCATTGCCCCACCT	TGGAAGGCCCAGATTC	123	[14]
TLR2 (Koala)	CCATTCCAAGTGAGGGGCAA	ACTCCAGTCAGCAAGGCAAG	122	KP792545.1
TLR3 (Koala)	GGAATGGCTTGGGTTGGAGT	AGCCACTGGAAAGAAAAATCATCT	162	KP792547.1
TLR4 (Koala)	TCCACAAGAGCCGGAAAGTC	GAGTTCCACCTGTTGCCGTA	176	KP792551.1
TLR5 (Koala)	CCTTAGCCTGGATGGCAACA	GGTAGGGTCAGGGGATAGCA	109	KP792550.1
TLR6 (Koala)	TTCAGTTTCCCGCCCAACTA	ATGTGGCCATCCACTTACCA	157	KP792540.1
TLR7 (Koala)	TTGCCTTGTAACGTCACCCA	GTGAGGGTCAGGTTGGTTGT	119	KP792558.1
TLR8 (Koala)	CCTCTTCGTTTACCACCCTCC	CTTCAAAGGCCCCGTCATCT	178	KP792567.1
TLR9 (Koala)	ATCTTCAGCCACTTCCGCTC	AGGCTCTCTCCAGCCCTAAA	133	KP792555.1
TLR10 (Koala)	GCCCTAAAGGTGGAGCATGT	TATATGTGGCATCCCCGCAC	123	KP792539.1
TLR13 (Koala)	AGCCTACTGGTGGCTATGGA	TGGCCAGGTACAGGGACTTA	172	KP334161.1

**Table 2 animals-11-00983-t002:** Koalas used in this study.

Koala	Age (at the Time of Sampling)	Sex	KoRV Subtypes		
			KoRV-A	KoRV-B	KoRV-C
KJ	6 y	Female	Positive	Negative	Negative
KY	2 y 4 m	Female	Positive	Negative	Negative
KYB	1 y	Female	Positive	Negative	Negative
KM	12 y	Female	Positive	Positive	Negative
K1	4 y 5 m	Male	Positive	Positive	Negative
K2	4 y 5 m	Female	Positive	Positive	Negative
H8	5 y 5 m	Female	Positive	Negative	Positive
KS	1 y 6 m	Male	Positive	Positive	Positive
H7	4 y 11 m	Female	Positive	Positive	Positive
H9	10 y 3 m	Female	Positive	Positive	Positive
Joey	6 m	Male	Positive	Negative	Positive

**Table 3 animals-11-00983-t003:** Hematological data from koalas.

Koala	Blood Parameters							Clinical Determination
	WBC (10^2^/μL)	RBC (10^4^/μL)	HGB (g/dL)	PCV (%)	MCV (fL)	MCH (pg)	MCHC (g/dL)	
KJ	125	359	13.2	37.8	105.3	36.8	34.9	Healthy
KY	62	342	12.4	37.6	109.9	36.3	33	Healthy
KYB	52	308	12	35.9	116.6	39	33.4	Healthy
KM	2275	241	8.8	25.4	105.4	36.5	34.6	Leukemic (subsequently died)
K1	1007.5	149	6	17	114.1	40.3	35.3	Leukemic
K2	77	256	7.9	38	148.4	30.9	20.8	Healthy
H8	82	342	13	38	111.1	38	34.2	Healthy
KS	4000	23.4	8.8	26	111.1	37.6	33.8	Leukemic
H7	97	330	12.2	35.3	107	37	34.6	Healthy
H9	52	319	11	33.8	106	34.5	32.5	Healthy

WBC, white blood cell; RBC, red blood cell; HGB, hemoglobin; PCV, packed cell volume; MCV, mean corpuscular volume; MCH, mean corpuscular hemoglobin; MCHC, mean corpuscular hemoglobin concentration. Except for koalas KY and H7, blood parameter data are from our recent study [12,38].

## Data Availability

Not applicable.

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
