# Peer review of "Toll-Like Receptor Expression Profiles in Koala (Phascolarctos cinereus) Peripheral Blood Mononuclear Cells Infected with Multiple KoRV Subtypes"

_animals, 2021, doi:10.3390/ani11040983_

Round 1

Reviewer 1 Report

This is an interesting subject and the better understanding on immune response in koalas is definitely of major relevance as it may provide support to understand their susceptibility to different diseases.

That being said, I think the present manuscript shows a lack of foundation to support this goal. I understand that working with wildlife and natural disease is almost never easy in terms of controlling conditions and ideal specimens. For that very reason, I guess that a clear presentation of what has been done is important for the understanding of the study. Below I present some specific comments.

2.1 Sample collection: Please, provide a clear breakdown of the animals against collected samples (Tissue Vs PBMC). At some stage, it is confusing how many animals were actually used, if 11 or 13, considering the joey and the one that died. The table clarifies some, but the information on the text needs to be clear to the reader.

Tables 1 and 2 should be part of the results, not of the methods.

Figure 1 would be better if presented on the methods, embedded in the item 2.1.  

3.2 How was the lymphoma diagnosed based on haematological examination? Wouldn’t this be leukaemia? Please, provide the criteria you used to consider lymphoma/leukaemia on the methods.

I do not see any comparison between the results of TLR + retrovirus and clinical/haematology. It would be interesting to hear about it, even if it is “no correlation”.

3.4 It is said the TLR7 was highly expressed in the bone marrow. How was this given a grade (highly)? What was this compared to?

Lines 256-261: This fraction would fit better in the introduction as part of the rational of the study.

The discussion needs to engage more with the results. I understand these are preliminary results, but a well-founded discussion will provide grounds for further studies. I suggest that a parallel is made with the role of TLR in the immune response to retrovirus in other species. Also, please, comment on the higher expression of certain TLR in KoRV-B-and KoRV-C-positive koalas. How does it correlate possibly with the pathogenesis? What is the role of these specific TLR in the antiviral/antiretroviral response?

For further investigation (Line 267), isn’t it also important to know the expression of TLR in retrovirus negative koalas?

Author Response

This is an interesting subject and the better understanding on immune response in koalas is definitely of major relevance as it may provide support to understand their susceptibility to different diseases.

That being said, I think the present manuscript shows a lack of foundation to support this goal. I understand that working with wildlife and natural disease is almost never easy in terms of controlling conditions and ideal specimens. For that very reason, I guess that a clear presentation of what has been done is important for the understanding of the study. Below I present some specific comments.

Response: We would like to thank the reviewers for their sincere comments.

2.1 Sample collection: Please, provide a clear breakdown of the animals against collected samples (Tissue Vs PBMC). At some stage, it is confusing how many animals were actually used, if 11 or 13, considering the joey and the one that died. The table clarifies some, but the information on the text needs to be clear to the reader.

Response: We regret for any confusion, and in line with the reviewer comment, we have clearly mentioned the total number of animals used in this study. In addition, we mentioned the animals separately that were used for blood and tissue collection (page 3, section 2.1).

Tables 1 and 2 should be part of the results, not of the methods.

Response: Thanks for the reviewer comments. In agreement with the reviewer comments, we have moved the Table 1 and 2 from methods to results section.

Figure 1 would be better if presented on the methods, embedded in the item 2.1.

Response: Following reviewer comments, we have presented Figure 1 in the item 2.1. under methods section.

3.2 How was the lymphoma diagnosed based on haematological examination? Wouldn’t this be leukaemia? Please, provide the criteria you used to consider lymphoma/leukaemia on the methods.

Response: Thanks for the reviewer comments. In line with reviewer comment we have modified Lymphoma to Leukemia, as this diagnosis was based on high white blood cell count during hematological examination.

I do not see any comparison between the results of TLR + retrovirus and clinical/haematology. It would be interesting to hear about it, even if it is “no correlation”.

Response: Thanks for the reviewer comments. We have updated the test as per the reviewer comment (Page 10, last paragraph before conclusions, Line 271-279).

3.4 It is said the TLR7 was highly expressed in the bone marrow. How was this given a grade (highly)? What was this compared to?

Response: Thanks for the reviewer comments. As per reviewer comment we have updated the text. A comparison was made among the TLRs measured, where TLR7 was found to be expressed most, compared to other TLRs in the bone marrow (Fig.3C).

Lines 256-261: This fraction would fit better in the introduction as part of the rational of the study.

Response: Agreeing with reviewer comment, we have moved the commented part to introduction (page 2 yellow highlighted part, Line 84-89).

The discussion needs to engage more with the results. I understand these are preliminary results, but a well-founded discussion will provide grounds for further studies. I suggest that a parallel is made with the role of TLR in the immune response to retrovirus in other species. Also, please, comment on the higher expression of certain TLR in KoRV-B-and KoRV-C-positive koalas. How does it correlate possibly with the pathogenesis? What is the role of these specific TLR in the antiviral/antiretroviral response?

Response: As per reviewer comments we have updated the text including role of TLR in immune response and pathogenesis in retrovirus infection in human and animals (Ref: 48, 49, 50). We have also commented on higher expression of certain TLRs that need to be investigated in the future study (Line 281-283).

For further investigation (Line 267), isn’t it also important to know the expression of TLR in retrovirus negative koalas?

Response: Thanks for the reviewer comments. We have updated the text as per the reviewer comment (Page 10, line 3).

Reviewer 2 Report

This work represents a significant contribution to our understanding of innate immune response in koalas with respect to KoRV infection.

Minor

Line 48: I wonder if listing nine references for KoRV association with neoplasia and chlamydial disease is not a bit too much.

Line 55: Do the authors have any reference for the claim that KoRV endogenization is still ongoing? Perhaps rewrite as “presumed” or ‘suspected”.

Table 1: Please reformat the table contents for clarity

Line 114: Remove “, and are shown as a reference.”

Do the authors have any thoughts on why most TLRs measured were expressed at relatively high levels in koala KM, a KoRV-B positive koala but not in the other two koalas, K1 and K2? I find this intriguing. Although the authors have not reported the viral load in these koalas, it might be a good idea for further studies to find an association, if any exists, between viral load and TLR mRNA expression.

Author Response

This work represents a significant contribution to our understanding of innate immune response in koalas with respect to KoRV infection.

Response: We would like to thank the reviewers for their sincere comments.

Minor

Line 48: I wonder if listing nine references for KoRV association with neoplasia and chlamydial disease is not a bit too much.

Response: Thanks for the reviewer comments. We realized same as the reviewer comment; however, to avoid any biasness in citing articles we preferred mentioning these references.

Line 55: Do the authors have any reference for the claim that KoRV endogenization is still ongoing? Perhaps rewrite as “presumed” or ‘suspected”.

Response: Thanks for the reviewer comments. As per reviewer comments we have rewritten the text (line 57).

Table 1: Please reformat the table contents for clarity

Response: Thanks for the reviewer comments. In line up with the reviewer comments we have reformatted the table for clarity.

Line 114: Remove “, and are shown as a reference.”

Response: Thanks for the reviewer comments. We have deleted the commented part.

Do the authors have any thoughts on why most TLRs measured were expressed at relatively high levels in koala KM, a KoRV-B positive koala but not in the other two koalas, K1 and K2? I find this intriguing. Although the authors have not reported the viral load in these koalas, it might be a good idea for further studies to find an association, if any exists, between viral load and TLR mRNA expression.

Response: Thanks for the reviewer comments. It is difficult to answer exactly at this stage, however the possibility of the age difference could not be ruled out, where koala KM was 12 years-old and K1 and K2 were 4 years 5 months-old.  However, it is of our interest to investigate the association in the future study, as reviewer commented.
